# Intermittent Theta Burst Stimulation Improves Motor and Behavioral Dysfunction through Modulation of NMDA Receptor Subunit Composition in Experimental Model of Parkinson’s Disease

**DOI:** 10.3390/cells12111525

**Published:** 2023-06-01

**Authors:** Milica Zeljkovic Jovanovic, Jelena Stanojevic, Ivana Stevanovic, Andjela Stekic, Samuel J. Bolland, Nebojsa Jasnic, Milica Ninkovic, Marina Zaric Kontic, Tihomir V. Ilic, Jennifer Rodger, Nadezda Nedeljkovic, Milorad Dragic

**Affiliations:** 1Laboratory for Neurobiology, Department for General Physiology and Biophysics, Faculty of Biology, University of Belgrade, 11000 Belgrade, Serbia; 2Institute for Medical Research, Military Medical Academy, 11000 Belgrade, Serbia; 3Medical Faculty of Military Medical Academy, University of Defense, 11000 Belgrade, Serbia; 4School of Biological Sciences, The University of Western Australia, Perth, WA 6009, Australia; 5Perron Institute for Neurological and Translational Science, Perth, WA 6009, Australia; 6Department for Comparative Physiology and Ecophysiology, Faculty of Biology, University of Belgrade, 11000 Belgrade, Serbia; 7Department of Molecular Biology and Endocrinology, Vinca Institute of Nuclear Sciences-National Institute of the Republic of Serbia, University of Belgrade, 11000 Belgrade, Serbia

**Keywords:** Parkinson’s disease, 6-OHDA, rTMS, iTBS, NMDA receptor, neuroprotection

## Abstract

Parkinson’s disease (PD) is the second most common neurodegenerative disorder characterized by the progressive degeneration of the dopaminergic system, leading to a variety of motor and nonmotor symptoms. The currently available symptomatic therapy loses efficacy over time, indicating the need for new therapeutic approaches. Repetitive transcranial magnetic stimulation (rTMS) has emerged as one of the potential candidates for PD therapy. Intermittent theta burst stimulation (iTBS), an excitatory protocol of rTMS, has been shown to be beneficial in several animal models of neurodegeneration, including PD. The aim of this study was to investigate the effects of prolonged iTBS on motor performance and behavior and the possible association with changes in the NMDAR subunit composition in the 6-hydroxydopamine (6-OHDA)-induced experimental model of PD. Two-month-old male Wistar rats were divided into four groups: controls, 6-OHDA rats, 6-OHDA + iTBS protocol (two times/day/three weeks) and the sham group. The therapeutic effect of iTBS was evaluated by examining motor coordination, balance, spontaneous forelimb use, exploratory behavior, anxiety-like, depressive/anhedonic-like behavior and short-term memory, histopathological changes and changes at the molecular level. We demonstrated the positive effects of iTBS at both motor and behavioral levels. In addition, the beneficial effects were reflected in reduced degeneration of dopaminergic neurons and a subsequent increase in the level of DA in the caudoputamen. Finally, iTBS altered protein expression and NMDAR subunit composition, suggesting a sustained effect. Applied early in the disease course, the iTBS protocol may be a promising candidate for early-stage PD therapy, affecting motor and nonmotor deficits.

## 1. Introduction

Parkinson’s disease (PD) is the second most common neurodegenerative disorder, affecting ~8 million people worldwide, with a prevalence increasing 2.5-fold over the past three decades [1]. The main characteristic of PD is the progressive degeneration of dopaminergic (DA) neurons in the *substantia nigra pars compacta* (SNpc) and the resulting decrease in striatal DA levels. The degeneration of the complex nigrostriatal networks and the resulting predominance of corticostriatal glutamatergic inputs seem to be crucial for the pathogenesis of PD [2,3,4] and responsible for most of the observed motor and behavioral dysfunction (i.e., cognitive impairment, anxiety, depression, sleep disorder) [5]. The currently available symptomatic therapy, a combination of levodopa/carbidopa, after an initial period of significant benefit, usually leads to drug-related motor complications (i.e., “wearing-off” symptoms, dyskinesia, “on-off” phenomenon), autonomic dysfunction and mood swings, as well as drug-related side effects (psychosis), all of which severely reduce quality of life [6]. Therefore, the development of new therapeutic approaches targeting all aspects of the disease remains the highest priority and an unmet need. Intracerebral injection of the neurotoxin 6-hydroxydopamine (6-OHDA) provides a valuable experimental rodent model that selectively targets DA neurons and induces neurodegeneration with synaptic rearrangements and motor and behavioral deficits consistent with those seen in humans [7,8,9]. The model reproduces the changes in the postsynaptic *N*-methyl-*D*-aspartate receptors (NMDAR) subunit composition and the resulting changes in basal ganglia connectivity typical of PD patients [3]. Notably, 6-OHDA–induced denervation alters relative GluN1 and GluN2B levels without affecting GluN2A [10,11,12], suggesting that variations in NMDAR subunit composition play a central role in deregulating corticostriatal plasticity [3,4]. Overall, the 6-OHDA-induced PD model is a reliable tool for testing the clinical efficacy of novel therapeutic approaches in PD.

Repetitive transcranial magnetic stimulation (rTMS) is a safe and noninvasive method of brain stimulation that has emerged as one of the potential candidates for PD therapy. In rodents, rTMS has been shown to be capable of eliciting complex neurobiochemical effects affecting early gene expression, changes in Ca^2+^ dynamics, changes in neurotransmitter release and glutamate receptor expression, reduction of oxidative stress and inflammation and activation of neurotrophic factors [13]. Previous research by several groups shows that intermittent theta burst stimulation (iTBS), a highly efficient excitatory protocol of rTMS, has beneficial effects on neuroinflammation, anxiety-like behavior, depressive-like behavior and learning and memory in several animal models of neurodegeneration [14,15,16]. To date, a few studies have been conducted to evaluate the effects of various rTMS protocols on the rodent model of PD [14,17]. Most studies reported beneficial effects of rTMS, but focused primarily on motor symptoms and survival of DA neurons (for a review please see [17]), neglecting behavioral aspects of the disease and underlying mechanisms. Given the complexity of PD pathology and the presence of nonmotor dysfunction that often precedes motor symptoms, further research is needed to provide a clear rationale for whether and how an rTMS-specific protocol, such as iTBS, can influence pathological processes underlying both motor and nonmotor aspects of PD. Therefore, the aim of this study was to examine the effects of prolonged iTBS on motor performance and behavior and the potential crosstalk with alterations in synaptic plasticity and NMDAR subunit composition. Our findings contribute to the understanding of the mechanisms underlying iTBS effects in experimental PD and provide new insights into how iTBS might be used as a therapy in PD patients.

## 2. Materials and Methods

### 2.1. Animals and Housing Conditions

A total of 90 two-month-old male Wistar rats (280 ± 20 g), bred at the Center of Veterinary Services animal facility, Ministry of Defense, were used for this study. Animals (3–4/cage) were maintained under constant conditions (23 ± 2 °C, 12 h light/dark cycle), a standard diet and tap water ad libitum. All experimental procedures were performed in accordance with the EU Directive 2010/63/EU and approved by the Ethics Committee for Animal Experiments of the College University of Belgrade—Faculty of Biology (No. 323-07-08250/2021-05). The experiments were repeated twice to avoid a litter effect (n = 50 per experiment).

### 2.2. Unilateral 6-Hydroxydopamine Lesion of the Right Substantia Nigra Pars Compacta

The animals were positioned in a stereotaxic frame (Stoetling Co., Wood Dale, IL, USA) under ketamine (100 mg/kg) and xylazine (10 mg/kg) anesthesia. Two microliters of 6-OHDA (6 μg/μL, Sigma Aldrich, St. Louis, MO, USA) in sterile saline supplemented with 0.2% ascorbic acid were injected into the region of rSNpc (right SNpc), whereas the same volume of vehicle was injected into the lSNpc (left SNpc) (+5.40 mm AP; ±2.10 mm ML and +7.40 mm DV, according to the stereotaxic Atlas of Paxinos and Watson). The neurotoxin was administered through a 50-µL Hamilton syringe at a constant flow rate of 0.4 µL/min (Harvard Apparatus, Holliston, MA, USA) [18]. The needle was left in place for an additional 5 min to allow diffusion of the solution in the SNpc and then slowly retracted. Another group of animals (control) received the vehicle solution bilaterally in the SNpc using the same procedure. Seven days postsurgery, a tail suspension test was performed to functionally assess the motor asymmetry induced by unilateral administration of 6-OHDA and performance in this test was used as a selection criterion [19]. In contrast, animals without motor asymmetry were excluded from the experiment (n = 4).

### 2.3. Experimental Design

Figure 1 summarizes the experimental design. Animals injected unilaterally with 6-OHDA were randomly divided into three experimental groups: 6-OHDA (n = 22), 6-OHDA + iTBS (n = 22) and 6-OHDA + iTBSsh (sham; n = 20). The fourth group of animals received the vehicle solution bilaterally and was used as controls (n = 20). The animals were subjected to the iTBS protocol or sham stimulation for 21 days and sacrificed after 30-dpi by decapitation (Harvard Apparatus, Holliston, MA, USA).

### 2.4. Theta Burst Stimulation Protocol

Seven days after surgery, animals in the 6-OHDA + iTBS group underwent the intermittent theta burst stimulation (iTBS) protocol, as previously reported [16]. Because DA neuronal cell death peaks approximately 7 days after surgery, coinciding with the onset of motor symptoms [20,21] the iTBS treatment began at this time point. Stimulation was performed with the MagStim Rapid^2^ device and a 25-mm figure-of-eight coil (MagStim Company, Whitland, UK). Twenty trains of ten bursts each (three pulses at a frequency of 50 Hz), repeated at 5 Hz (10 s intervals between trains, with a total duration of the stimulation of 192 s, with the magnetic stimulation intensity set at 35%) were delivered to each animal. The animals were gently held during stimulation while allowed to move freely during the 10 s intervals between trains. The sham group (6-OHDA+ iTBS sham) was exposed to the noise artifact by placing a cage containing two animals next to the stimulation device. The same treatment protocol was repeated for 21 consecutive days. The 6-OHDA and control groups did not receive iTBS treatment.

### 2.5. FEM Modeling Methodology

To investigate the TMS parameters and induced electric field (E) in this study, a 3D finite element method (FEM) was used to create a rat head and TMS model. The head model was derived from high-resolution anatomical T2-weighted images with isotropic voxels of an ex vivo rat’s head (weighing 280 ± 20 g) acquired with a 9.4T Bruker Biospec 94/30 small animal MRI machine. MRI images were processed and high-resolution images [22] were registered using the FMRIB software library [23] and the original rat brain was extracted using the Brain Extraction Tool (BET) (Figure 2). The 3D surfaces for each tissue type were segmented using ITK-SNAP [24] based on the image signal intensity values and imported into COMSOL Multiphysics version 6.0 for optimization, meshing and TMS simulation (Appendix A). The final rat head and coil model consisted of ~1.13 million domain elements. The brain was 26.85 mm long (excluding the spinal trigeminal tract and spinal cord). The MagStim Rapid2 TMS device and 25-mm eight-coil models were created using two homogenized multiturn coils (inner diameter 18 mm, outer diameter 42 mm), each with 14 turns of apartment copper (0.75 mm × 6 mm) [25,26]. The model boundary contained an infinite domain unaffected by boundary conditions. All model domains were meshed with at least one “extra fine” element size and the coil and its core were meshed with the swept function. rTMS was simulated by driving the coil with a 584.5 V (V_c_) pulse generated by the MagStim system, with the current change (dI/dt) inducing the E-field peaking at 50.7 A/µs (dI/dt|_Max_ = V_c_/L). This simulation was performed in the frequency domain using the biphasic pulse rise time similar to the method used by Tang et al. (2016) [27]. The coil was aligned centrally over the rat brain so that the current direction of the coil would induce an E-field in the brain running anterior to posterior. Dielectric properties such as isotropic conductances (σ) were set for each tissue layer as soft tissue = 0.465 S/m, skull = 0.02 S/m, cerebrospinal fluid = 1.654 S/m, gray matter = 0.106/m and white matter = 0.126 S/m; other dielectric properties were set to values used in previous rodent TMS studies [28,29]. The E-field and magnetic flux density (B) induced by rTMS were calculated using the magnetic and electric field equations in the AC /DC module of COMSOL, which solves Maxwell’s equations by determining the magnetic vector potential field (A) in the frequency domain. The rate of change of the coil voltage determines the electric field induced in the brain and the magnetic flux density is given by the expressions E = −∇V − *j*ωA and B = ∇ × A, respectively, where ∇ is the curl, ω is the frequency and *j* is the free current density.

### 2.6. Behavioral Tests

Behavioral tests were performed in a secluded room to provide acoustic and visual isolation. Before each test, animals were allowed at least 60-min habituation period. A single researcher conducted all behavioral tests to ensure consistency in treatment of the animals. Odors were removed by cleaning the apparatus with 70% alcohol between test sessions. Control and treated animals were treated in parallel on the same day to keep conditions as constant as possible.

### 2.7. Rota-Rod Test

A rota-rod device (Elunit, Belgrade, Serbia) was used to assess motor coordination and balance. Seven days before surgery, the animals underwent three conditioning sessions on a rota-rod device. The animals were placed on a stationary suspended cylinder (rod) for 30 s to stimulate fall avoidance behavior. Animals were then placed on the rotating cylinder at a constant speed of 10 rpm for 90 s. Animals that did not withstand the challenge in all training sessions were removed from the experiment. The experimental animals were subjected to 3 rota-bar test sessions, accelerating from 4 to 20 rpm in 200 s [30]. Each test session consisted of three trials on the rota-rod, with a maximum duration of 200 s per trial and a 30 min inter-trial interval. For each animal, the latency to fall and traveled distance were recorded and the best performance from the trial was used for further analysis.

### 2.8. Limb Use Asymmetry (Cylinder Test)

The cylinder test is used to study spontaneous use of the forelimbs [31]. An animal was placed in a glass cylinder (H: 40 cm × D: 20 cm) and the number of times it reared up and touched the cylinder wall was recorded for 5 min. Wall contacts were sorted as contralateral forelimb (CF), ipsilateral forelimb (IF) or both forelimb (BF) contacts. The number of impaired forelimb contacts relative to the total number of contacts was calculated using [(CF × 0.5 BF)/(IF + CF + BF)] and was used as an index of asymmetry [32]. Unaware of the experimental groups, two researchers independently analyzed the data and averaged the results.

### 2.9. Open Field

After completion of iTBS treatment, an open field test (OFT) was performed to assess anxiety-like behavior [16,33]. The animals were placed in the upper right corner of a black arena (100 × 100 × 50 cm) divided into 25 × 25 cm squares and their activity was recorded for 5 min. The total number of movements and the time spent in the center fields (in seconds) were analyzed using ANY-maze Video Tracking System 7.11.

### 2.10. Object Recognition Test

Object recognition was applied to test the learning and memory performance of the 6-OHDA animals subjected to iTBS. The animals were placed in the center of the arena, equidistant from two identical rectangular objects. They were allowed to freely explore the arena and the objects for 5 min (sampling phase) before returning to their home cages. After one hour, the animals returned to the arena, where a new conical object was introduced in place of a rectangular one. Their activity was recorded for an additional 5 min (test phase), provided that sniffing, climbing and exploring the object for more than 2 s was classified as active exploration. Performance was analyzed by ANY-maze Video Tracking System 7.11. The time spent with the novel object relative to the total time spent with both objects is expressed as the recognition index (RI) [34].

### 2.11. Sucrose Preference Test

The sucrose preference test was used to assess anhedonia and/or depressive-like behavior [35]. Briefly, animals were deprived of food and water for 18 h, beginning at 3 pm the day before the test. At 9 am, the animals were placed in a cage for one hour and given two preweighed bottles, one containing tap water and the other containing 2% sucrose. The procedure was repeated for three consecutive days, switching the position of the bottles. The volume of ingested liquids was measured after each session and the mean volume was used to evaluate the sucrose preference (%) = [(sucrose intake/(sucrose intake + water intake)) × 100].

### 2.12. Brain Tissue Preparation and Immunohistochemical Staining

The brains were rapidly removed from the skull after decapitation (n = 3–4/group) and fixed in 4% paraformaldehyde (PFA) for 24 h, cryoprotected and dehydrated in graded sucrose solution (10–30% in 0.2 M PBS, pH 7.4). The 25 µm-thick coronal sections at the level of the caudoputamen and the midbrain were mounted on supefrost glass slides, air-dried for 1–2 h at RT and stored at −20 °C until use.

After rehydration in PBS, the sections were treated with 0.3% hydrogen peroxide for 20 min and washed with PBS for 3 × 5 min. Subsequently, sections were blocked with 5% normal donkey serum at room temperature for 1 h, followed by incubation with primary antibodies overnight at 4 °C (Table 1). The slides were then probed with appropriate secondary antibodies (Table 1) for 2 h at room temperature. The signal was visualized using the 3,3′-S-diaminobenzidine-tetrahydrochloride kit (DAB, Abcam, Cambridge, UK) as a chromogen for HRP-conjugated secondary antibodies. After dehydration in graded ethanol (70–100%) and clearance in xylene, the sections were mounted with the DPX-mounting medium (Sigma Aldrich, USA). The sections were examined under a LEITZ DM RB light microscope (Leica Mikroskopie and Systems GmbH, Wetzlar, Germany) equipped with a LEICA DFC320 CCD camera (Leica Microsystems Ltd., Heerbrugg, Switzerland) and analyzed using LEICA DFC Twain Software (Leica, Wetzlar, Germany).

### 2.13. Measurements of Dopamine (DA) Content in Striatum by HPLC Assay

For the HPLC analysis, brains (5–6 animals/experimental group) were isolated from the skull and the midbrain and striatum were dissected for subsequent determination of dopamine and serotonin concentrations [36]. Tissue samples were homogenized in DEPROT (1 mg/10 µL) using an Ultra-Turrax homogenizer, sonicated (3 × 10 s), centrifuged (30 min, 15,000 rpm, 4 °C) and the supernatants were transferred to separate tubes. An aliquot of each sample (40 µL) was injected into the UltiMate3000 HPLC system (Thermo Scientific, Waltham, MA, USA) and applied to C18 HPLC column (Thermo Scientific, Waltham, MA, USA) with 100 mM ammonium formate buffer, pH 3.6 (A) and methanol (B), as a mobile phase. The mobile phase was pumped at a flow rate of 500 µL/min, with an initial A:B ratio of 98:2%. Under these conditions, serotonin and dopamine were readily separated and detected by an electrochemical detector (850 mV, 25 °C). Data were analyzed using the Chromeleon7 Chromatography Data System (Thermo Scientific). The catecholamine concentrations were expressed as µg/mg tissue.

### 2.14. Tissue Isolation and Western Blot Analysis

Animals in the Sham group and the iTBS group (5–7 animals per group) were decapitated and the brains were rinsed in ice-cold saline. The right caudoputamen (rCPu) and left caudoputamen (lCPu) and right midbrain (rMB) and left midbrain (lMB) were dissected and separately frozen in liquid nitrogen and stored at −80 °C. For Western blot analysis, the expression level of target proteins in the lCPu and lMB of each animal was used as an internal control for the expression level in the rCPu and rMB, respectively. Tissue samples were homogenized in the isolation buffer (0.32 M sucrose, 10 mM HEPES, pH 7.4) at 4 °C, the resulting homogenates were centrifuged at 3000× *g* for 10 min at 4 °C and the supernatants obtained were collected for Western blot analysis. The protein content in each sample was determined using the Pierce™ BCA Protein Assay Kit (Thermo Fisher Scientific, USA). Equal sample aliquots (20 µg of the sample proteins) were resolved using SDS-PAGE and transferred to PVDF membrane using semidry transfer, as described previously [37]. The supporting membrane was incubated with primary antibodies (Table 1), rinsed in TBST and incubated with appropriate horseradish-peroxidase (HRP)-conjugated secondary antibodies using SmartBlot apparatus. Chemiluminescent signals were detected by the ECL solution (Bio-Rad, Hercules, CA, USA) in ChemiDocIt Imager (UltraViolet Products Ltd., Cambridge, UK). The optical densities (OD) of the target band and GAPDH band (loading control) in each lane were determined in the ImageJ program (https://imagej.nih.gov/ij/, accessed on 1 October 2020) and the ratio in each lane was expressed relative to the same ratio in Sham-L. The results are expressed as mean ± SD, from n = 2–4 independent replicates.

### 2.15. Statistical Analysis

All data were analyzed for normality using the Shapiro—Wilk test and appropriate parametric or nonparametric tests were used. The results of the behavioral tests were analyzed using one-way ANOVA, followed by Tukey’s *post hoc* test for multiple comparisons. The results obtained by HPLC and Western blot analysis were analyzed using Student’s *t* test or the Mann—Whitney test if the normality condition was not met. The values represent mean ± SD as indicated in Figure legends. The values of *p* < 0.05 were considered statistically significant. Analysis and graphical presentation were performed in the GraphPad Prism 9.0 (San Diego, CA, USA) software package.

## 3. Results

### 3.1. FEM Modeling Results

FEM modeling of the MagStim Rapid^2^ apparatus estimated the generated B-field peak to be 1.28 T at the base of each coil forming the figure of eight loop. At the superior surface of the brain (closest to the coil), the value was estimated to be 698.5 mT, whereas in the CPu it ranged from 480.4 mT to 302.2 mT and in the SN from 358.8 mT to 303.4 mT (Figure 2B). Accordingly, the estimated E-field value at the cortical surface of the brain was 124.05 V/m, decreasing down to a minimum of ~21 V/m at the base of the brain (Figure 2B–D). The CPu was located between 5.24 mm and 9.81 mm below the scalp surface and experienced simulated E-field values between 57.92 V/m and ~33 V/m at these depths, respectively. The SN was located 8.95 mm to 10.87 mm below the scalp surface and exhibited E-field values between ~34 V/m and 37.98 V/m, respectively, (Figure 2A,B).

### 3.2. Intermittent Theta Burst Stimulation Improves 6-OHDA-Induced Motor Dysfunction

The effects of 6-OHDA injection into the rSNpc and the iTBS protocol on the extent of motor impairment were determined by measuring latency to fall and traveled distance in the rota-rod test (Figure 3). In each of the three consecutive sessions, the latency to fall was significantly lower in the 6-OHDA group compared to the control animals, whereas the animals receiving iTBS stimulation showed a dramatic improvement in motor performance, reflected as significantly prolonged latency to fall, which was comparable to the control (Figure 3A—F_(3, 76)_ = 21.10, *p* < 0.0001; Figure 3B—F_(3, 76)_ = 17.52, *p* < 0.0001; Figure 3C—F_(3, 76)_ = 13.89, *p* < 0.0001). The same holds for travelled distance measures, which were significantly reduced in 6-OHDA animals and comparable to the control level in the iTBS group (Figure 3D—F_(3, 76)_ = 19.01, *p* < 0.0001; Figure 3E—F_(3, 76)_ = 17.26, *p* < 0.0001; Figure 3F—F_(3, 76)_ = 14.84, *p* < 0.0001). The effect of iTBS on the motor performance was further tested in the cylinder test. The use of contralateral forelimb was severely impaired in the 6-OHDA animals compared to the control. The animals subjected to iTBS showed a significant improvement, reflected in the percentage of contralateral forelimb contacts comparable to the control (Figure 3G, F_(3, 76)_ = 17.36, *p* < 0.0001).

### 3.3. Intermittent Theta Burst Stimulation Improves 6-OHDA-Induced Non-Motor Symptoms and Neurochemical Imbalance

The efficacy of iTBS in attenuating 6-OHDA-induced anxiety-like behavior was examined in the open-field test. Measured parameters, including the number of entries (Figure 4A, F_(3, 75)_ = 11.20, *p* < 0.0001) and time spent in central fields (Figure 4B, F_(3, 73)_ = 21.05, *p* < 0.0001), show significant improvement in iTBS-treated animals compared to the 6-OHDA group (Figure 4A,B). Similarly, iTBS attenuated depressive-like behavior in the sucrose preference test (F_(3, 40)_ = 11.49, *p* < 0.0001). The sham and 6-OHDA animals showed a moderate reduction in sucrose intake compared to the control, whereas sucrose intake in iTBS animals was comparable to the control (Figure 4C). The efficacy of iTBS stimulation on learning and memory was tested through an object recognition test (Figure 4D). A significant difference was observed in the recognition index between the iTBS, 6-OHDA and sham groups (F _(3, 62)_ = 16.97, *p* < 0.0006), which implies a beneficial influence of iTBS on short-term memory performance.

The neurochemical changes induced by 6-OHDA and iTBS stimulation were assessed by determining the difference between the rCPu and lCPu in respect of protein abundance of dopamine transporter (DAT) (Figure 4E) and dopamine and serotonin levels (Figure 4F,G). The protein abundance of DAT protein expression in the iTBS group was lower than in the sham group (*t* = 3.082, *p* < 0.01). The ratio of dopamine (Figure 4F, *t* = 2.590, *p* < 0.05) and serotonin (Figure 4G, *t* = 3.798, *p* < 0.01) in the rCPu and lCPu increased significantly in the iTBS animals compared to the sham group.

### 3.4. Intermittent Theta Burst Stimulation Reduces 6-OHDA-Induced Neuronal Death of the Lesioned SNpc and Loss of TH Positive Fibers Density in the Caudoputamen

The neuroprotective effects of iTBS treatment on dopaminergic neurons in the SNpc and their projections in the CPu were assessed using tyrosine hydroxylase (TH) immunohistochemistry (Figure 5A,B) as well as TH protein quantification with Western blot analysis (Figure 5C,D). Unilateral injection of 6-OHDA in a sham group resulted in a marked decrease in TH-immunoreactivity in rSNpc (Figure 5A) and TH-positive projections in the rCPu (Figure 5B) compared to the contralateral sides. Analysis of relative protein expression of TH-positive neurons (Figure 5C, *t* = 5.471, *p* < 0.001) in MB and TH-positive fibers (Figure 5D, *t* = 3.147, *p* < 0.01) in the CPu confirmed the higher survival of the iTBS group compared to the sham group.

### 3.5. Intermittent Theta Burst Stimulation Modulates NMDA Receptor Subunit Composition

The expression levels of the NMDA receptor subunits GluN1, GluN2A and GluN2B, the AMPA receptor (GluR1) and the glutamate transporters GLAST (EAAT1) and GLT (EAAT2) were further examined in the CPu after 6-OHDA and iTBS treatment. The expression levels of BDNF, SIN and PSD-95, which are involved in synaptic plasticity and functional recovery were also examined (Figure 6). As presented in Figure 5, protein abundance of GluN1 (Figure 6A, *t* = 5.040, *p* < 0.001), GluN2A (Figure 6B, *t* = 5.537, *p* < 0.001), GLAST (Figure 6D, *t* = 4.212, *p* < 0.001), GLT1 (Figure 6E, *t* = 2.783, *p* < 0.05) and BDNF (Figure 6F, *t* = 3.672, *p* < 0.006) in rSNpc compared to lSNpc significantly increased in animals receiving iTBS treatment relative to the sham group. The iTBS treatment resulted in an increase in SYN/PSD-95 ratio relative to the sham group (Figure 6H, *t* = 4.768, *p* < 0.001).

## 4. Discussion

The aim of the present study was to investigate the effect of a prolonged iTBS stimulation protocol on gross motor performance and nonmotor symptoms in the 6-OHDA model of Parkinson’s disease. Neurodegeneration was induced by unilateral injection of 6-OHDA into the right SNpc, resulting in selective DA neuronal cell loss and anterograde degeneration of the nigrostriatal neural pathway. We would like to emphasize that the experimental paradigm used in this work may be more appropriate than the other commonly used when the toxin is infused in the CPu. Although both experimental paradigms lead to similar motor impairments, the underlying cause might be different at the molecular level. Because 6-OHDA enters the cell via DAT, when used in CPu it unselectively destroys all synaptic terminals and possibly glial cells expressing DAT [38,39] as well as other elements expressing serotonin and norepinephrine transporters, since 6-OHDA can also enter the cell via these transporters [38]. On the other hand, if it is injected into the SNpc, it destroys the cell bodies of the dopaminergic neurons and consequently all projections to the CPu, but also to other regions [40] leading to a more suitable model of PD. The main focus of this paper was to evaluate the specific iTBS protocol that started 7 dpi with the onset of motor and behavioral deficits [19,20,41] and lasted for three consecutive weeks. The effects of iTBS were assessed in terms of motor and nonmotor symptoms and underlying neurochemical and biochemical responses.

To analyze which parts of the brain are affected by the iTBS protocol, we created the FEM model. The FEM model describes that E-field levels in most parts of the brain, including the caudoputamen and substantia nigra, are above the value of ~28 V/m, sufficient to generate action potentials in the affected neuronal tissue [42], predicting that the iTBS protocol would have been effective in eliciting electrical changes in the structures studied. It should be noted that the electrical field generated also affects glial cells and their physiology, which may contribute to the overall response seen after iTBS [43]. The FEM model gives us the opportunity and rationale to hypothesize that the iTBS protocol used in this study may have a direct effect by targeting the CPu and SNpc, apart from the likely indirect effects resulting from stimulation of other brain regions.

Previous modeling attempts have described that TMS-induced maximal E-field values depend strongly on the species and brain size. However, rodent and human values are similar for the 25-mm coil used in our study, supporting the translational potential of this protocol [44].

The study confirmed that the iTBS protocol was able to improve the motor deficits of 6-OHDA-affected animals, as evidenced by significantly improved performance on the rota-rod and cylinder tests. Motor improvements were observed as early as one week after initiation of iTBS stimulation, in contrast to other rTMS protocols that required two or more weeks of application to produce measurable motor progress [45,46]. Furthermore, the iTBS protocol resulted in sustained motor improvement that lasted for at least three weeks after intoxication; that is, for the entire duration of the stimulation. At the cellular level, motor improvement may be associated with a significant reduction in DA neuronal cell loss and preservation of nigrostriatal projections [14]. Several TMS protocols have been shown to reduce neuronal cell death by interfering with pro-apoptotic signaling pathways [14,16] and reducing the production of reactive oxygen species [47] involved in 6-OHDA-induced neurotoxicity [48]. In addition to direct neuroprotective effects, iTBS also attenuates glial cell-mediated secondary damage [15,49], all of which may be responsible for the increased TH expression observed after treatment.

Our study demonstrated that the iTBS protocol improves nonmotor symptoms, including anxiety- and depressive/anhedonic-like behavior and cognitive deficits induced by 6-OHDA. The neurotoxin directly induces loss of dopaminergic neurons and subsequent dopamine depletion in the striatum, which contributes to both motor impairments and anxiety- and depressive/anhedonic-like behavior. Several studies have shown that 6-OHDA-induced loss of dopaminergic terminals, as well as other catecholaminergic projections, leads to a neurochemical imbalance that partially underlies the observed behavioral deficits [50]. Accordingly, iTBS protocol restored dopamine levels in the striatum, confirming previous data obtained with other rTMS protocols in animal models of neurodegenerative [20,46] and psychiatric disorders [51]. The recovery of dopamine levels after iTBS [20,43,46] is likely due to reduced loss of DA neuronal projections and striatal DAT expression. The beneficial effects of iTBS on behavioral and motor deficits in 6-OHDA are probably due, in part, to increased serotonin levels as well, which is critically involved in cognitive and motor functions and in the pathophysiology of depressive-like behavior in animals and humans [50,52]. Serotonin in the striatum originates from the dense terminal projections arising from the dorsal *raphe* nucleus. Its interactions with the striatal dopaminergic system [53] decrease corticostriatal [54] and thalamostriatal [53] glutamatergic input and modulated reward-mediated learning [55], thus the observed increase may influence aforementioned processes. The behavioral deficits resulting from 6-OHDA, such as anxiety- and depressive/anhedonic behavior, are a complex phenomenon with an elusive molecular background, so that increases in dopamine and serotonin, although beneficial, are not solely responsible for the deficits or improvements after iTBS. It is also worth noting that due to the size of the coil, the stimulation affects virtually the entire brain, so it is possible that some of the improvement in motor and nonmotor symptoms is also due to stimulation of other brain regions and increased hemispheric compensation [56]; however, this is beyond the scope of this study and requires further investigation.

In this regard, our study is the first to show changes in NMDAR subunit composition in a model of 6-OHDA-induced SNpc degeneration. In 6-OHDA animals, increased extracellular glutamate from corticostriatal inputs potentiates GluN1/GluN2B-mediated signaling, which, together with depletion of DA, can lead to motor and behavioral deficits [57]. The decreased expression of GluN1 and GluN2A and upregulation of GluN2B suggest that 6-OHDA alters NMDA receptor subunit composition in favor of GluN1 and GluN2B. Therefore, iTBS treatment reversed these effects and resulted in an increase in GluN1 and GluN2A and a decrease in GluN2B subunit expression. Interestingly, selective GluN2B receptor antagonism was found to alleviate symptoms in rodent and primate models of PD [58,59,60,61,62], suggesting an important role of GluN2B-mediated signaling in PD. Moreover, the iTBS-induced increase in GLAST and GLT1 expression may offset the increased extracellular glutamate levels. Several studies have shown that extracellular glutamate levels and/or binding of glutamate to its receptors are increased in both the 6-OHDA model and PD pathology [63,64,65]. Because chronic rTMS has been shown to induce the release of neurotransmitters in vivo, including glutamate, the increase in astrocytic GLAST and GLT1 may be a compensatory mechanism that can regulate extracellular glutamate levels, which is also relevant to the pathology.

Overall, our data suggest that iTBS enhances GluN1/GluN2A-mediated signaling and DA-glutamate crosstalk in the striatum, which is essential for motor and nonmotor behavior. Moreover, it has been shown that induction of LTP by high-frequency stimulation in the dorsolateral striatum requires GluN2A-, but not GluN2B-containing NMDAR [66]. Additional evidence comes from our findings that prolonged iTBS increased expression of both presynaptic and postsynaptic markers, which may indicate enhanced synaptic contacts and which may point to improved structural plasticity and improved motor/behavioral function [12,13]. This phenomenon was demonstrated by other groups in the PD model after acute iTBS [67,68]; however, it would require additional electrophysiological evidence to examine synaptic plasticity in our conditions. Dopaminergic terminals converge with cortical glutamatergic inputs on striatal spiny GABAergic neurons, which control many behavioral outputs [3]. NMDARs containing different subunits play distinct roles in the striatum, i.e., GluN2A-containing NMDARs regulate glutamatergic synaptic transmission and evoked dopamine release in the striatum [69]. Moreover, altered NMDAR subunit composition in the striatum has been shown to be closely associated with the pathophysiology and progression of both PD and experimental parkinsonism [4,10]. Finally, iTBS as well as other rTMS protocols have been shown to affect the expression of NMDAR subunits [70] and that their effects are mediated by NDMA signaling [71,72,73], further supporting the data we obtained.

In summary, although the results of the present study clearly demonstrate the beneficial effects of iTBS and address some of the potential underlying mechanisms, several limitations of the study should be noted. The first relates to the technical limitations associated with the size and manual placement of the coil, which do not allow focal stimulation of specific areas but can be considered as whole-brain stimulation. Therefore, the observed effects of iTBS may be the result of both cortical and subcortical stimulation and their interconnectivity. The second and a very important limitation concerns the nature of the tissue component. More specifically, it is impossible to precisely determine the changes in a particular cellular compartment (i.e., extrasynaptic vs. synaptic, membrane vs. cytoplasm) in the fraction used, so we can only discuss the overall changes in the striatal region. Finally, more in-depth analyses are required to connect the changes in NMDAR components to the behavioral deficits to strengthen and elucidate the observed benefits following iTBS.

## 5. Conclusions

To our knowledge, this is the first study to show positive effects of prolonged iTBS on motor and especially, on emotional behavior as well as on learning and memory in the 6-OHDA-induced SNpc degeneration experimental paradigm of PD. This study is also the first to report molecular changes that may contribute to the understanding of the action of iTBS in this model. Overall, the results suggest that prolonged iTBS rescues dopaminergic cells and increases striatal levels of DA, serotonin and glutamate transporter expression and alters NMDAR subunit composition, leading to predominant GluN1/GluN2A-mediated signaling. In conclusion, iTBS protocol, if applied at the onset of early symptoms, may be a promising candidate for the early-stage therapy of PD targeting motor and nonmotor deficits.

## Figures and Tables

**Figure 1 cells-12-01525-f001:**
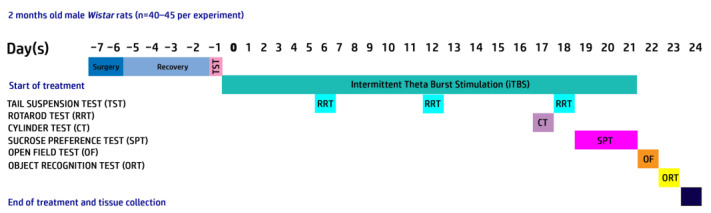
Outline of the experimental paradigm.

**Figure 2 cells-12-01525-f002:**
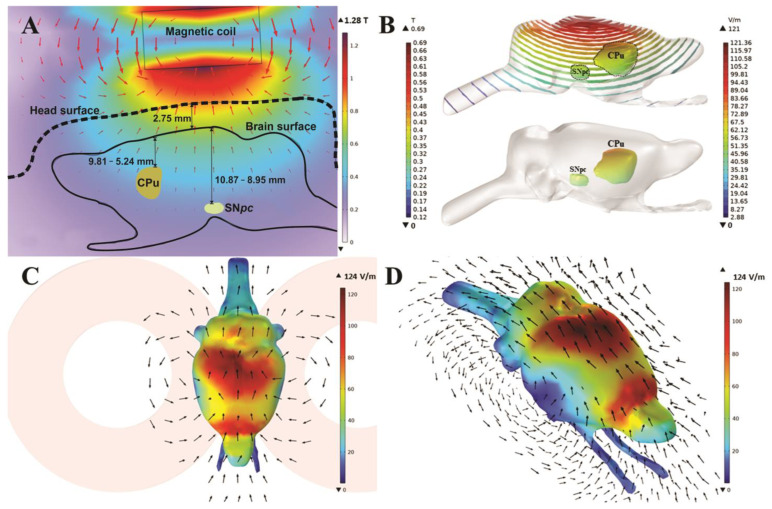
Heat maps of magnetic field (**A**,**B**) and electric field (**C**,**D**) measured in the stimulated iTBS FEM model. Representative FEM models of iTBS stimulation depicting 2D (**A**) and 3D (**B**) heat maps of magnetic field and electric fields (**C**,**D**). Arrows show the direction of the generated electric field (**C**,**D**). Scale bars = 10 mm.

**Figure 3 cells-12-01525-f003:**
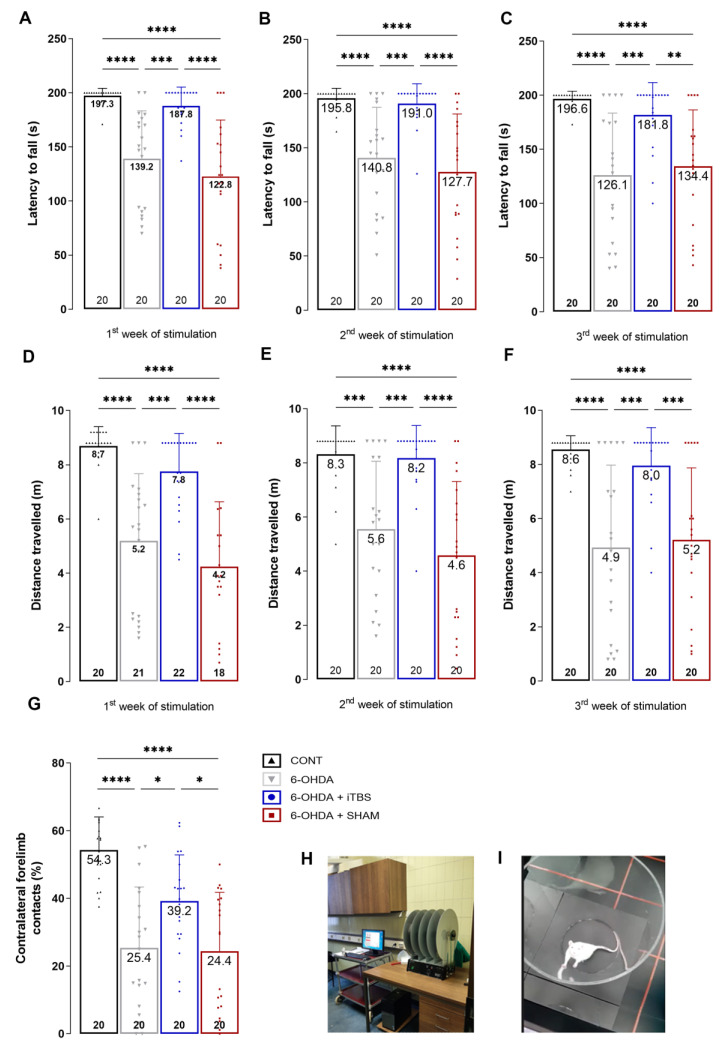
Intermittent theta burst stimulation improves 6-OHDA-induced motor dysfunction. Motor performance in control and 6-OHDA-lesioned rats following iTBS treatment was evaluated using rota-rod (**A**–**F**) and cylinder test. Rota-rod measured the latency to fall (s) and travelled distance (m) after first, (**A**,**D**) second (**B**,**E**) and third (**C**,**F**) week of stimulation. Cylinder test measured the contralateral forelimb contacts with the wall (**G**). Values are expressed as mean ± SD. Results of *post hoc* Tukey’s test and significance are shown inside the graphs: * *p* < 0.05, ** *p* < 0.01, *** *p* < 0.001, **** *p* < 0.0001. Dots in the graphs represent the values of individual animals. (**H**) Rota-rod setting. (**I**) Cylinder arena.

**Figure 4 cells-12-01525-f004:**
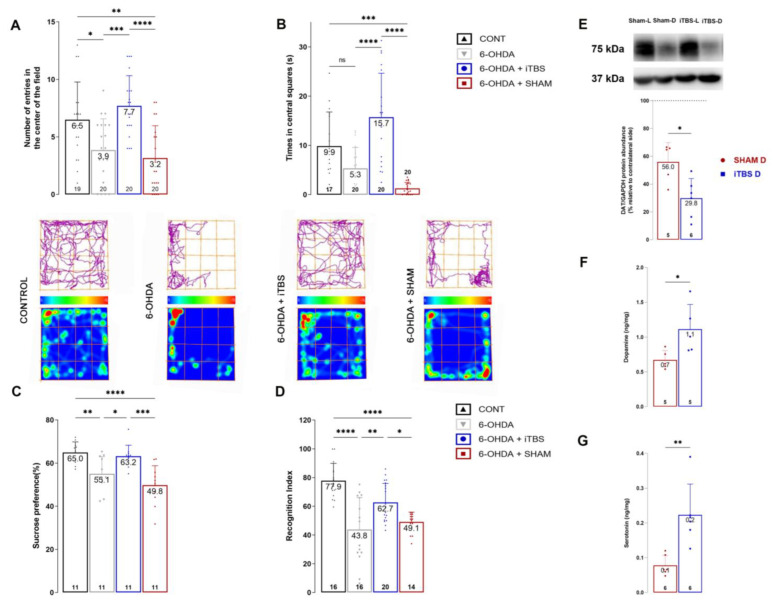
Intermittent theta burst stimulation improves 6-OHDA-induced nonmotor symptoms and neurochemical imbalance. Quantitative analysis of open field behavioral test represented as number of center entries (**A**) and time in central squares (**B**). Track plots and heat maps of animals from each group are shown below the graphs. Sucrose preference test was used to assess anhedonia (**C**), while short-term memory was assessed with novel object recognition test expressed as recognition index (**D**). Results of *post hoc* Tukey’s test and significance are shown in graphs, * *p* < 0.05, ** *p* < 0.01, *** *p* < 0.001, **** *p* < 0.0001, ns—not significant. Representative immunoblot membrane and quantitative data analysis showing relative DAT protein abundance in caudoputamen (**E**). Bars represent the mean value of target protein normalized to GAPDH abundance ± SD (from n = 5–6 individual animals, in 2–4 technical replicates), expressed relative to left caudoputamen, which serves as internal control, arbitrarily defined as 100%. Results of HPLC analysis of dopamine (**F**) and serotonin (**G**) concentrations in lesioned caudoputamen expressed as ng/mg tissue. Data are expressed as mean ± SD (n = 5–6/group). * *p* < 0.05 different from sham (two-tailed unpaired Student’s *t*-test, **E**–**G**). Dots in the graphs represent values of individual animals.

**Figure 5 cells-12-01525-f005:**
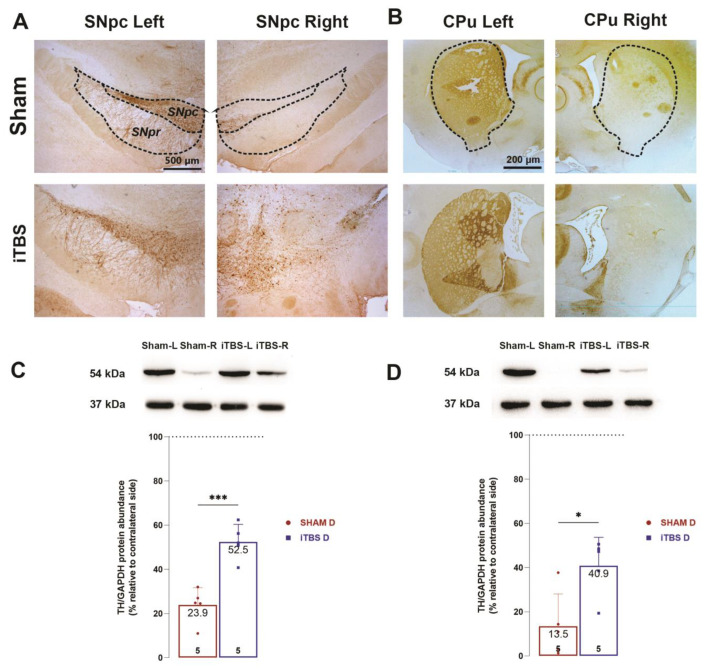
Intermittent theta burst stimulation reduces 6-OHDA-induced neuronal death of the lesioned SNpc and loss of TH positive fibers density in the caudoputamen. Representative coronal sections of TH-positive neurons in the SNpc (**A**) and TH-positive fibers in striatum (**B**) from sham-treated and iTBS-treated rats after 4 weeks. Scale bar: 500 µm. Representative immunoblot membrane and quantitative data analysis showing relative TH protein abundance in SNpc (**C**) and caudoputamen (**D**). Bars represent the mean value of target protein normalized to GAPDH abundance ± SEM (from n = 5 individual animals, in 2–4 technical replicates), expressed relative to left midbrain or caudoputamen, which serves as internal control, arbitrarily defined as 100%. Results of two-tailed unpaired Student’s *t*-test and significance are shown in graphs: * *p* < 0.05, *** *p* < 0.001.

**Figure 6 cells-12-01525-f006:**
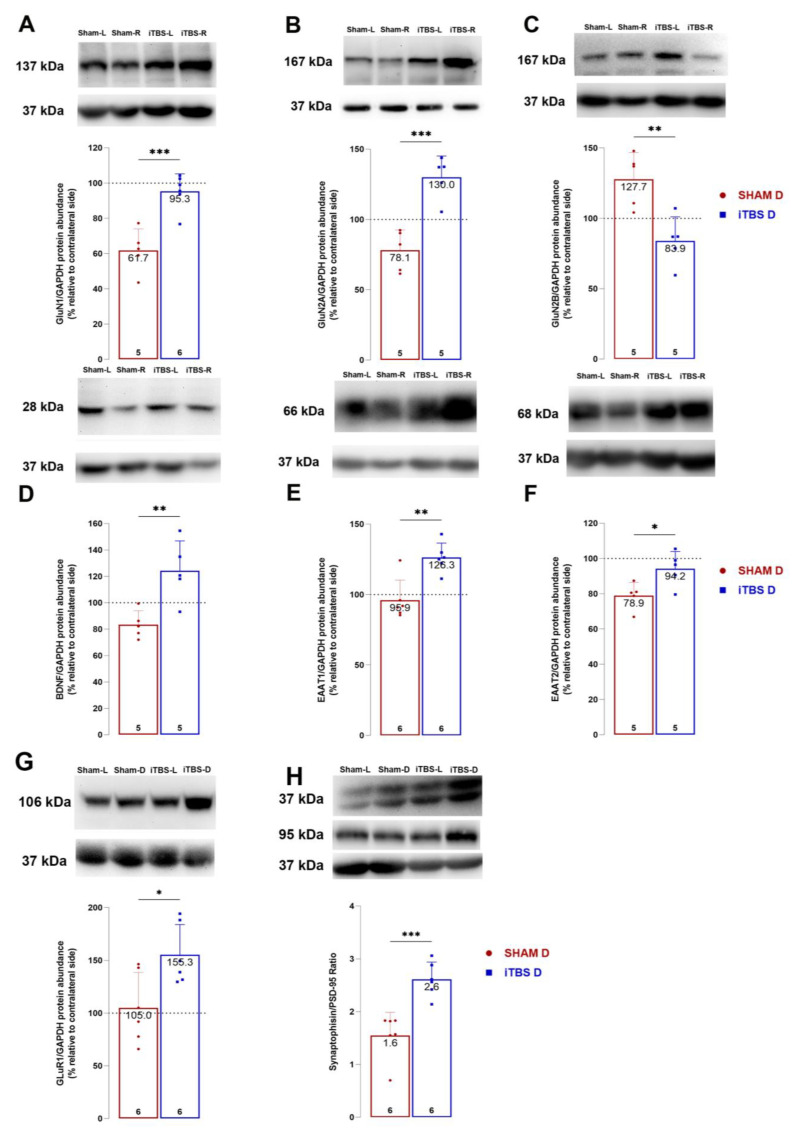
Intermittent theta burst stimulation modulates NMDA receptor subunit composition. Representative immunoblot membrane and quantitative data analysis showing relative subunit protein abundance of GluN1 (**A**), GluN2A (**B**), GluN2B (**C**), BDNF (**D**), EAAT1 (**E**), EAAT2 (**F**) and GluR1 (**G**). Relative abundance of the ratio of synaptic proteins Synaptophysin and PSD-95 ratio in caudoputamen of sham and iTBS animals (**H**). Bars represent the mean value of target protein normalized to GAPDH abundance ± SD (from n = 5–6 individual animals, in 2–4 technical replicates), expressed relative to left caudoputamen, which serves as internal control, arbitrarily defined as 100%. Results of two-tailed unpaired Student’s *t*-test and significance are shown in graphs: * *p* < 0.05, ** *p* < 0.01, *** *p* < 0.001.

**Table 1 cells-12-01525-t001:** List of used primary and secondary antibodies.

Antibody	Source and Type	Used Dilution	Manufacturer
DAT	Rabbit, polyclonal	1:500 ^WB^	Abcam, #ab184451, RRID:AB_2890225
TH	Rabbit, polyclonal	1:2000 ^WB^, 1:500 ^IHC^	Millipore #AB152, RRID:AB_390204
GluN1	Rabbit, polyclonal	1:4000 ^WB^	Cell Signaling Technology, #5704, RRID:AB_1904067
GluN2A	Rabbit, polyclonal	1:4000 ^WB^	Millipore, #07-632, RRID:AB_310837
GluN2B	Mouse, monoclonal	1:3000 ^WB^	Abcam #ab93610, RRID:AB_10561972
BDNF	Goat, polyclonal	1:1000 ^WB^	Santa Cruz Biotechnology, #sc-33904, RRID:AB_2259044
EAAT1	Rabbit, polyclonal	1:1000 ^WB^	Cell Signaling Technology, #5684T, RRID:AB_10695722
EAAT2	Rabbit, polyclonal	1:1000 ^WB^	Abcam, #ab69098, RRID:AB_2190732
GluR1	Mouse, monoclonal	1:1000 ^WB^	Santa Cruz Biotechnology, #sc-55509, RRID:AB_629532
Synaptophysin	Rabbit, polyclonal	1:5000 ^WB^	Santa Cruz Biotechnology, #sc-9116, RRID:AB_2199007
PSD-95	Mouse, monoclonal	1:1000 ^WB^	Millipore, #MAB1598, RRID:AB_94278
GAPDH	Rabbit, polyclonal	1:2000 ^WB^	Thermo Fisher Scientific, #PA1-987, RRID:AB_2107311
Goat anti-rabbit IgG, HRP-conjugated	Goat, polyclonal	1:30,000 ^WB^	Abcam, #ab6721, RRID: AB_955447
Goat anti-mouse IgG, HRP-conjugated	Goat, polyclonal	1:30,000 ^WB^	Abcam, #ab97240, RRID:AB_10695944
Rabbit anti-goat IgG, HRP-conjugated	Rabbit, polyclonal	1:10,000 ^WB^	R and D Systems, #HAF017, RRID:AB_562588

WB—western blot; IHC—immunohistochemistry.

## Data Availability

The data presented in this study are available on request from the corresponding author. The data are not publicly available due to policy of our Institute.

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
