# Peer review of "Intermittent Theta Burst Stimulation Improves Motor and Behavioral Dysfunction through Modulation of NMDA Receptor Subunit Composition in Experimental Model of Parkinson’s Disease"

_cells, 2023, doi:10.3390/cells12111525_

Round 1
Reviewer 1 Report (Previous Reviewer 2)
Cells – Manuscript ID: cells-2320100 – “Intermittent Theta Burst Stimulation Improves Motor and
Behavioral Dysfunction Through Modulation of NMDA Receptor Subunit Composition in Experimental
Model of Parkinson’s Disease”, by Milica Zeljkovic Jovanovic, Jelena Stanojevic, Ivana Stevanovic, Andjela
Stekic, Samuel J Bolland, Nebojsa Jasnic, Milica Ninkovic, Marina Zaric Kontic, Tihomir V Ilic, Jennifer
Rodger, Nadezda Nedeljkovic and Milorad Dragic
=
An improved version of the previous Manuscript ID: cells-2213447 – “Intermittent Theta Burst
Stimulation Improves Motor and Behavioral Dysfunction Through Modulation of NMDA Receptor Subunit
Composition in Experimental Model of Parkinson’s Disease”, by Milica Zeljkovic Jovanovic, Andjela Stekic,
Nadezda Nedeljkovic, Milorad Dragic, Jelena Stanojevic , Ivana Stevanovic, Milica Ninkovic, Tihomir V
Ilic, Samuel J Bolland, Nebojsa Jasnic , Marina Zaric Kontic and Jennifer Rodger.
In the first version of the manuscript, authors report beneficial effects of prolonged (three consecutive
weeks) intermittent theta burst stimulation (iTBS) on motor and emotional behavior and on learning
and memory in the 6-OHDA experimental rat model of PD. They also report molecular and cellular
changes.
The authors submitted a carefully-prepared revision, which satisfactorily addressed the remaining
concerns.
They have addressed all comments.
Author Response
Thank you for your valuable comments.
Reviewer 2 Report (Previous Reviewer 1)
This is a potentially interesting paper that is aimed at describing the chronic effects of transcranial magnetic intermittent theta-burst stimulation (iTBS). The Authors report therapeutic effects on both motor and nonmotor symptoms in rats that received intracerebral injection of 6-hydroxydopamine (6-OHDA) as an experimental model of parkinsonism. The Authors attributed these beneficial effects on changes in the levels of NMDA and AMPA receptors subunits in striatal samples.
The paper is generally well written and clear, even if the use of a computational model is not clear.
Beside this, I have a couple of doubts that blunt my enthusiasm for the manuscript in the current form but that can be easily addressed maybe without adding new experiments.
1. The Authors aim to demonstrate that iTBS can change the function of glutamatergic receptors without providing a real neuronal marker of this change. They measured NMDA and AMPA receptors subunits levels in the striatum but it is not clear if the change observed is related to an acute effect or a chronic effect. Thus, is it not clear if this effect might last over time, Did the Authors tried, in their experience, to decipher this aspect? This might provide an important result in support of a possible long term plasticity that might involve morphological changes, as claimed by the Authors.
2. The data on glial changes are interesting. Also here, the changes in glutamate transporters in astrocytes are a key finding, but it is not clear if this might have actually an impact in the striatal synapses. Are there specific papers in the literature showing increased release of glutamate with chronic TMS treatments, like with in vivo mocrodialysis?
Did the authors investigate microglia phenotype? Since microglia is activated earlier in response to damage and is able to engage astrocytes it would be interesting that besides the GFAP marker the authors could provide information about other microglial markes to show if there is a shift between proinflammatory and anti-inflammatory phenotypes. This is not a trivial aspect as the treatment is initiated when the inflammatory response in the nigrostriatal system is supposed to be very high and neuroinflammatory markers have an impact on the re-organization of the synaptic transmission in both the striatal and in the mesencephalic dopaminergic neurons.
All these information either provided with new set of data or discussed through the evidence already existent in the literature, would strengthen the authors conclusions that iTBS reduce non motor symptoms of parkinsonism. In fact these aspects that include complex autonomic, sensorial, psychiatric signs are not easily modelled in laboratory animals with this toxin-based model and, without building a solid ground, the conclusions do not result convincing.
Author Response
We would like to thank the Reviewer for critically assessing our manuscript and for raising concerns and pointing to issues that if resolved would significantly improve the clarity and quality of the paper.
We have addressed all comments point by point from the Reviewer and introduced the necessary changes in the manuscript. Our responses are in red.
The paper is generally well written and clear, even if the use of a computational model is not clear.
Thank you very much for your comment. We use the computer model to predict whether, given that the output is set to a submotor threshold, the applied protocol can reach the structures of interest (i.e., caudoputamen and substantia nigra pars compacta), and if the stimulation reaches the structures, is it able to induce electrical changes that can affect neuronal status. This may provide a significant data when discussing the results, given that we may postulate that part of the effects we observe may be due to direct stimulation of the structures, and not only because cortex or other structures are stimulated as well.
Beside this, I have a couple of doubts that blunt my enthusiasm for the manuscript in the current form but that can be easily addressed maybe without adding new experiments.
- The Authors aim to demonstrate that iTBS can change the function of glutamatergic receptors without providing a real neuronal marker of this change. They measured NMDA and AMPA receptors subunits levels in the striatum but it is not clear if the change observed is related to an acute effect or a chronic effect. Thus, is it not clear if this effect might last over time, Did the Authors tried, in their experience, to decipher this aspect? This might provide an important result in support of a possible long term plasticity that might involve morphological changes, as claimed by the Authors.
Thank you very much for your comment. It seems that the effect is present at least for the duration of the stimulation, as we performed a Rota-Rod test after each week of stimulation. This highlights the fact that the effect does not change as long as the stimulation continues. There is some data in the literature that there is an aftereffect of stimulation at many levels, from the molecular to the behavioral, however, these changes are protocol-based and there is much data on human subjects (see review). Furthermore, it is clear from the literature that iTBS/rTMS exerts its effects on NMDA subunit receptor expression (see PMID: 25172625, 28550530) and that some of its effects are mediated by NMDA signaling (see PMID: 23197741, 24656783, 32289670, 24342462, 10224311). It is a very interesting question how long iTBS lasts under our conditions after termination of stimulation, and it certainly requires a new research direction.
- The data on glial changes are interesting. Also here, the changes in glutamate transporters in astrocytes are a key finding, but it is not clear if this might have actually an impact in the striatal synapses. Are there specific papers in the literature showing increased release of glutamate with chronic TMS treatments, like with in vivo mocrodialysis?
Thank you for your question. There are studies which have demonstrated that various protocols of rTMS affects the levels on neurotransmitters including glutamate, dopamine, serotonin and norepinephrine assessed by in vivo microdialysis (please see PMID 11029641, 32210744, 12213264, 15187982) and by other methods as well. Furthermore, the expression of EAAT1/2 transporter may be put in perspective because several studies demonstrated that in 6-OHDA model, there is an increase of glutamate release from corticostriatal inputs (please see 22936308), but it should be noted that there are conflicting results in this regard.
Did the authors investigate microglia phenotype? Since microglia is activated earlier in response to damage and is able to engage astrocytes it would be interesting that besides the GFAP marker the authors could provide information about other microglial markes to show if there is a shift between proinflammatory and anti-inflammatory phenotypes. This is not a trivial aspect as the treatment is initiated when the inflammatory response in the nigrostriatal system is supposed to be very high and neuroinflammatory markers have an impact on the re-organization of the synaptic transmission in both the striatal and in the mesencephalic dopaminergic neurons.
Thank you very much for your question. We added the figure regarding glial status only in response to Reviewer because it is part of our currently ongoing project in which we are investigating the role of glial cells and purinergic signaling in this experimental model after iTBS as a function of time. We have strong evidence that the phenotype of microglia is shifting toward a more favorable phenotype for tissue repair/remodeling and that it is exerting more markers of phagocytosis and generally has anti-inflammatory phenotype, which has been demonstrated in other models (please see PMID 35504201).
All these information either provided with new set of data or discussed through the evidence already existent in the literature, would strengthen the authors conclusions that iTBS reduce non motor symptoms of parkinsonism. In fact these aspects that include complex autonomic, sensorial, psychiatric signs are not easily modelled in laboratory animals with this toxin-based model and, without building a solid ground, the conclusions do not result convincing.
Thank you very much for your comment. We have addressed all of the above points and included them in the discussion of the revised manuscript. It is true that modeling nonmotor symptoms in laboratory animals is a real challenge because much of the symptoms observed in patients with PD are due to higher cognitive functions-i.e., it is postulated that many of the PD patients develop anxiety because they are aware of the disease progression and uncertainty about the effects of treatment, etc. However, some non-motor symptoms are exclusively due to a neurochemical imbalance resulting from the depletion of dopamine and other catecholamines in the 6-OHDA-induced model and confirmed in the literature (please see some of the references PMID: 31056008, 32458386, 31141725). Therefore, we have included a segment in the discussion that provides the basis for assuming that the nonmotor symptoms are a consequence of the neurochemical imbalance and that the observed improvements are partly due to changes in this segment that we have demonstrated.
Round 2
Reviewer 1 Report (Previous Reviewer 2)
Comments for authors
Cells – Manuscript ID: cells-2320100 – “Intermittent Theta Burst Stimulation Improves Motor and Behavioral Dysfunction Through Modulation of NMDA Receptor Subunit Composition in Experimental Model of Parkinson’s Disease”, by Milica Zeljkovic Jovanovic, Jelena Stanojevic, Ivana Stevanovic, Andjela Stekic, Samuel J Bolland, Nebojsa Jasnic, Milica Ninkovic, Marina Zaric Kontic, Tihomir V Ilic, Jennifer Rodger, Nadezda Nedeljkovic and Milorad Dragic
=
An improved version of the previous Manuscript ID: cells-2320100 – “Intermittent Theta Burst Stimulation Improves Motor and Behavioral Dysfunction Through Modulation of NMDA Receptor Subunit Composition in Experimental Model of Parkinson’s Disease”, by Milica Zeljkovic Jovanovic, Jelena Stanojevic, Ivana Stevanovic, Andjela Stekic, Samuel J Bolland, Nebojsa Jasnic, Milica Ninkovic, Marina Zaric Kontic, Tihomir V Ilic, Jennifer Rodger, Nadezda Nedeljkovic , Milorad Dragic.
In the previous version of the manuscript, authors report beneficial effects of prolonged (three consecutive weeks) intermittent theta burst stimulation (iTBS) on motor and emotional behavior and on learning and memory in the 6-OHDA experimental rat model of PD. They also report molecular and cellular changes.
The authors submitted a carefully-prepared revision, which satisfactorily addressed the remaining concerns.
They have addressed all comments.
Therefore, the reviewer recommends publication.

Author Response
Thank you for your comments.
Reviewer 2 Report (Previous Reviewer 1)
General comments
The paper is generally well written and clear, even if the use of a computational model is not clear.
Thank you very much for your comment. We use the computer model to predict whether, given that the output is set to a submotor threshold, the applied protocol can reach the structures of interest (i.e., caudoputamen and substantia nigra pars compacta), and if the stimulation reaches the structures, is it able to induce electrical changes that can affect neuronal status. This may provide a significant data when discussing the results, given that we may postulate that part of the effects we observe may be due to direct stimulation of the structures, and not only because cortex or other structures are stimulated as well.
It is more clear now, but I still feel that the computer model is not tied up with the rest of the story, also because there is no “return” to the predicted values or no validation of the estimation by the measure of some functional parameter, like the amplitude of a postsynaptic potential. Therefore, such additional results appear useless since they are not linked to a measurable result that can validate it. It appears like a pending issue.
Beside this, I have a couple of doubts that blunt my enthusiasm for the manuscript in the current form but that can be easily addressed maybe without adding new experiments.
1. The Authors aim to demonstrate that iTBS can change the function of glutamatergic receptors without providing a real neuronal marker of this change. They measured NMDA and AMPA receptors subunits levels in the striatum but it is not clear if the change observed is related to an acute effect or a chronic effect. Thus, is it not clear if this effect might last over time, Did the Authors tried, in their experience, to decipher this aspect? This might provide an important result in support of a possible long term plasticity that might involve morphological changes, as claimed by the Authors.
Thank you very much for your comment. It seems that the effect is present at least for the duration of the stimulation, as we performed a Rota-Rod test after each week of stimulation. This highlights the fact that the effect does not change as long as the stimulation continues. There is some data in the literature that there is an aftereffect of stimulation at many levels, from the molecular to the behavioral, however, these changes are protocol-based and there is much data on human subjects (see review). Furthermore, it is clear from the literature that iTBS/rTMS exerts its effects on NMDA subunit receptor expression (see PMID: 25172625, 28550530) and that some of its effects are mediated by NMDA signaling (see PMID: 23197741 slice cultures, 24656783, 32289670, 24342462, 10224311). It is a very interesting question how long iTBS lasts under our conditions after termination of stimulation, and it certainly requires a new research direction.
I appreciate this discussion, but citations should be carefully revised. The authors say that the effects are protocol-dependent. For example, it is true that rTMS with high-frequency stimulation is analogous to iTBS and I can accept to use a paper with HF rTMS to support data on iTBS. However, reference n. 68 (PMID 25172625) does not support the actual data for two reasons. First, Liu et al., use LIMs (a far more different protocol). Second, in that paper there is no direct evidence of changes in NMDAR subunits. This citation must be also removed from line 514 of the manuscript file.
2. The data on glial changes are interesting. Also here, the changes in glutamate transporters in astrocytes are a key finding, but it is not clear if this might have actually an impact in the striatal synapses. Are there specific papers in the literature showing increased release of glutamate with chronic TMS treatments, like with in vivo microdialysis?
Thank you for your question. There are studies which have demonstrated that various protocols of rTMS affects the levels on neurotransmitters including glutamate, dopamine, serotonin and norepinephrine assessed by in vivo microdialysis (please see PMID 11029641, 32210744, 12213264, 15187982) and by other methods as well. Furthermore, the expression of EAAT1/2 transporter may be put in perspective because several studies demonstrated that in 6-OHDA model, there is an increase of glutamate release from corticostriatal inputs (please see 22936308), but it should be noted that there are conflicting results in this regard.
This is an interesting elaboration that, since the paper is lacking functional data in support of the conclusions, must find its place in the discussion.
3. Did the authors investigate microglia phenotype? Since microglia is activated earlier in response to damage and is able to engage astrocytes it would be interesting that besides the GFAP marker the authors could provide information about other microglial markes to show if there is a shift between proinflammatory and anti-inflammatory phenotypes. This is not a trivial aspect as the treatment is initiated when the inflammatory response in the nigrostriatal system is supposed to be very high and neuroinflammatory markers have an impact on the re-organization of the synaptic transmission in both the striatal and in the mesencephalic dopaminergic neurons.
Thank you very much for your question. We added the figure regarding glial status only in response to Reviewer because it is part of our currently ongoing project in which we are investigating the role of glial cells and purinergic signaling in this experimental model after iTBS as a function of time. We have strong evidence that the phenotype of microglia is shifting toward a more favorable phenotype for tissue repair/remodeling and that it is exerting more markers of phagocytosis and generally has anti-inflammatory phenotype, which has been demonstrated in other models (please see PMID 35504201).
I am fine with this reply.
All this information either provided with new set of data or discussed through the evidence already existent in the literature, would strengthen the authors conclusions that iTBS reduce non motor symptoms of parkinsonism. In fact these aspects that include complex autonomic, sensorial, psychiatric signs are not easily modelled in laboratory animals with this toxin-based model and, without building a solid ground, the conclusions do not result convincing.
Thank you very much for your comment. We have addressed all of the above points and included them in the discussion of the revised manuscript. It is true that modeling nonmotor symptoms in laboratory animals is a real challenge because much of the symptoms observed in patients with PD are due to higher cognitive functions-i.e., it is postulated that many of the PD patients develop anxiety because they are aware of the disease progression and uncertainty about the effects of treatment, etc. However, some non-motor symptoms are exclusively due to a neurochemical imbalance resulting from the depletion of dopamine and other catecholamines in the 6-OHDA-induced model and confirmed in the literature (please see some of the references PMID: 31056008, 32458386, 31141725). Therefore, we have included a segment in the discussion that provides the basis for assuming that the nonmotor symptoms are a consequence of the neurochemical imbalance and that the observed improvements are partly due to changes in this segment that we have demonstrated.
The fact that the authors did not follow the suggestion of providing a new set of data is understandable. However, they did not either follow the suggestion of addressing these issues in the discussion, but I think it is mandatory to do that in order to strengthen the conclusions (except for the actual point 3 about glial data, which is a topic that they are developing for another paper and I respect this decision).
My suggestions were motivated by the fact that the paper has limits that are still not fully recognized in the last lines of the discussion. One of these is the claim of linking all-together the molecular effects of TMS to the behavioral outcome observed, with a reductionist approach.
Regarding the last reply, I think that the monoaminergic theory of depression is much more than that described by the authors here, and it has been firmly questioned since the discovery of antidepressant drug delayed effects. Therefore, a note of caution should be added by partially remodeling the sentence: “Several studies have shown that 6-OHDA-induced loss of dopaminergic terminals, as well as other catecholaminergic projections, leads to a neurochemical imbalance that partially underlies the observed behavioral deficits (49).”
Author Response
We would like to thank the Reviewer for careful consideration of the manuscript and for the comments which further introduced clarity and objectivity to the paper.
1) It is more clear now, but I still feel that the computer model is not tied up with the rest of the story, also because there is no “return” to the predicted values or no validation of the estimation by the measure of some functional parameter, like the amplitude of a postsynaptic potential. Therefore, such additional results appear useless since they are not linked to a measurable result that can validate it. It appears like a pending issue.
Thank you very much for your comment. Using this model should only confirm that the magnetic field can reach our structures, as SNpc and CPu are located deep in the brain, and that the magnetic field is sufficient to generate the electric field and affect both neurons and glial cells. There is a lot of variety in rTMS, and many protocols change the maximum power and duration, the magnetic coils vary greatly from lab to lab, often they are handmade, so we cannot fully rely on the data of others as their protocol may produce different strengths of magnetic/electric field. In addition, the available data support a translational potential, as the values for a 25-mm coil are similar in humans and rats. Therefore, we used it to calculate the strength and it is not the 'primary' result of our manuscript. We agree that it is an open question that is not fully related to the rest of the manuscript, but it provides a good predictive model for this and future research and can provide confidence to other researchers if they decide to use this protocol, and we and others can build on it in the future. In addition, we deleted Figure 2 (MRI images) because it does not provide useful information to the reader and may add unnecessary burden to the manuscript; therefore, we moved it to the supplemental material.
2) I appreciate this discussion, but citations should be carefully revised. The authors say that the effects are protocol-dependent. For example, it is true that rTMS with high-frequency stimulation is analogous to iTBS and I can accept to use a paper with HF rTMS to support data on iTBS. However, reference n. 68 (PMID 25172625) does not support the actual data for two reasons. First, Liu et al., use LIMs (a far more different protocol). Second, in that paper there is no direct evidence of changes in NMDAR subunits. This citation must be also removed from line 514 of the manuscript file.
Thank you for your comment and suggestion. We will remove the citation from the text.
3) This is an interesting elaboration that, since the paper is lacking functional data in support of the conclusions, must find its place in the discussion.
Thank you for the suggestion. We have added it to the discussion.
4) The fact that the authors did not follow the suggestion of providing a new set of data is understandable. However, they did not either follow the suggestion of addressing these issues in the discussion, but I think it is mandatory to do that in order to strengthen the conclusions (except for the actual point 3 about glial data, which is a topic that they are developing for another paper and I respect this decision).
My suggestions were motivated by the fact that the paper has limits that are still not fully recognized in the last lines of the discussion. One of these is the claim of linking all-together the molecular effects of TMS to the behavioral outcome observed, with a reductionist approach.
Regarding the last reply, I think that the monoaminergic theory of depression is much more than that described by the authors here, and it has been firmly questioned since the discovery of antidepressant drug delayed effects. Therefore, a note of caution should be added by partially remodeling the sentence: “Several studies have shown that 6-OHDA-induced loss of dopaminergic terminals, as well as other catecholaminergic projections, leads to a neurochemical imbalance that partially underlies the observed behavioral deficits (49).”
Thank you for your comment. This is a really interesting topic for discussion, especially with regard to the monoaminergic theory of depression. In an excellent paper published in Molecular Psychiatry, a meta-analysis was done looking at the causality behind SSIR prescription and serotonin levels in the brains of depressed patients, and it was found that depressed patients do not actually have less serotonin or that the receptors for serotonin have a lower binding capacity ( please see 35854107). On the other hand, rats infused with 6-OHDA have a significant reduction in all catecholamines, which affect various aspects of behavior such as motivation-driven behavior, hedonic behavior, and so on. Moreover, dopaminergic deafferentation affects virtually the entire brain, which receives projections of SNpc/SNpr, leading to changes in overall behavior, particularly 'catecholaminergic behavior." We agree that we cannot simply infer that the increase in serotonin and dopamine is the cause of the behavioral improvement, but we would like to leave open the possibility that the increase in 5 HT and DA mediated by iTBS could underlie some of the improvements, being cautious and acknowledging that it is a very complex matter that cannot be simply reduced to two molecules. Therefore, we softened our conclusion regarding molecular effects and added a section to the discussion that leaves the possibility open but acknowledges the complexity of the issue. Finally, we changed the term depressive-like to depressive/anhedonic-like behavior given that we performed only one behavioral test, and to be more certain which phenotype is produced we need more than one behavioral test.
This manuscript is a resubmission of an earlier submission. The following is a list of the peer review reports and author responses from that submission.
Round 1
Reviewer 1 Report
The goal of this paper is to demonstrate that chronic treatment with transcranial magnetic stimulation using an Intermittent Theta Burst Stimulation (iTBS) protocol has therapeutic effects on the motor and nonmotor symptoms in rats injected with 6-hydroxydopamine (6-OHDA) as an experimental model of parkinsonism and that these beneficial effects depend on alterations in synaptic plasticity and related AMPA and NMDAR subunit expression.
The most important limitation of this study is that the authors base their rationale on the possibility of demonstrating changes in synaptic plasticity, but neither electrophysiological nor morphological aspects of synaptic plasticity are investigated here.
Concerning the morphological changes, the authors acknowledge that the “observed changes in striatal plasticity are tightly connected to morphological changes in striatal neurons, which is a key structural component of underlying synaptic plasticity". However, the authors rely on previous papers showing that iTBS improves connectivity without providing themselves evidence: "Loss of dendritic spines and altered morphology of spiny GABAergic neurons in lesioned striatum (63) was shown to be reversed after acute iTBS treatment (60)”. One of these citations refers to a paper that used different stimulation (acute and not chronic, as in the current paper, ref 60). The other one, by Solis et al., investigated the morphological changes after 4 weeks without noninvasive stimulation. As in that important study (ref 63), the authors should provide morphological evidence that structural changes are observed in response to chronic iTBS starting at an early stage after the surgery (see below). This would be another way to support their conclusions.
Another critical aspect regards the molecular characterization of glutamatergic receptors and, in particular, the aim of linking the changes in postsynaptic density composition to the effects of TMS on emotional behavior. It is recognized that the most assessed method to get insights into the NMDAR subunit composition in relation to their role in the striatal synaptic plasticity (i.e., their surface expression or phosphorylation state) is the analysis of postsynaptic fractions that was not done here. Homogenates include all the membranes in the tissue and, therefore, provide information that is not specific to corticostriatal synapses.
As a result, the discussion presents several weaknesses regarding this point that cannot really be revised without reconsidering the possibility of adding new experiments. In fact, even if the authors recognize the limit of using homogenates, they still discuss their data, affirming that “iTBS positively altered NMDAR synaptic plasticity, suggesting a sustained effect”, a fact that cannot be demonstrated without providing further evidence.
The other critical limit is the study of the effects of iTBS on emotional behavior, as the relevant areas involved in regulating such behaviors are not explicitly investigated. Several leading research groups in the field of noninvasive stimulation have already tried to investigate non-motor aspects of disease in animal models, and the consensus is that it is too speculative to attribute a net change in emotions and cognition generated by a whole-brain stimulation to such a specific regional (striatum) change. Most of the studies, in fact, limited their report to motor activity by studying the dorsolateral striatum or motor cortices, and some good papers have investigated the possible underlying anti-inflammatory and anti-apoptotic actions of TMS. Besides, the neuroinflammatory and apoptotic response must also be analyzed here, as the time point for starting the treatment is very close to the surgery (7 days), generating an acute inflammation state rather than an established parkinsonian condition. Given that pro-inflammatory cytokines dramatically affect synaptic plasticity expression - particularly LTP, which is the form of synaptic plasticity that primarily involves the activation of GluN subunits - it is almost impractical to investigate plastic changes at this stage unless the goal is to study apoptotic cell death.
In the conclusions, the authors state that their study is the first to report molecular changes that could underpin the mechanism of action of iTBS in this model. However, other important papers reported molecular changes related to the parkinsonian condition using this model (Yang et al., Neuroreport 2010; Wang et al., J Neurosci 2011, and others reviewed in Uzair et al., 2022, also cited in this paper).
In commenting on their findings' impact, the authors state that “if the iTBS protocol is applied at the onset of early symptoms, it may be a promising candidate for effective early-stage therapy PD, targeting motor and nonmotor deficits”. However, a toxic model like 6-OHDA injection into the SNpc is not gradual at all and it is less suitable to study early symptoms of PD than other available models that would help to characterize the effects of iTBS on behavioral alterations often preceding motor symptoms. Therefore, in the current version, this paper does not “provide new insight into how iTBS could be exploited as a therapy in PD patients”.
The usefulness of a FEM 3D model is not introduced and is not further elaborated in the rest of the paper. Figure 3 seems to suggest that the stimulation has broad effects on the brain.
The manuscript text might benefit from a revision by a native English speaker.
Reviewer 2 Report
Comments for authors
Cells – Manuscript ID: cells-2213447 – “Intermittent Theta Burst Stimulation Improves Motor and Behavioral Dysfunction Through Modulation of NMDA Receptor Subunit Composition in Experimental Model of Parkinson’s Disease”, by Milica Zeljkovic Jovanovic, Andjela Stekic, Nadezda Nedeljkovic, Milorad Dragic, Jelena Stanojevic , Ivana Stevanovic, Milica Ninkovic, Tihomir V Ilic, Samuel J Bolland, Nebojsa Jasnic , Marina Zaric Kontic and Jennifer Rodger.
In this manuscript, the authors investigate the effects of prolonged (three consecutive weeks) intermittent theta burst stimulation (iTBS) on motor performance and behavior and the possible association with changes in synaptic plasticity and NMDAR subunit composition in the 6-hydroxydopamine (6-OHDA)-induced experimental rat model of Parkinson’s disease (PD). Authors describe beneficial effects of prolonged iTBS on motor and emotional behavior and on learning and memory in the 6-OHDA experimental rat model of PD. They also report molecular change. They show that prolonged iTBS rescues dopaminergic cells and increases striatal levels of DA, serotonin and expression of glutamate transporters and alters NMDAR subunit composition, leading to prevalent GluN1/GluN2A-mediated signaling.
This is an interesting paper addressing an interesting perspective candidate for the efficient therapy in early-stage PD targeting of motor and non-motor deficits.
Methods are relevant. They aren’t major flaws or biases and conclusions are based on the data. The literature is up-to-date. Then, the topic is suited to Cells.
Authors report precisely author contributions, ethics board approval, disclosure of funding and conflicts of interest. There isn’t reason to suspect research misconduct.
Discussion and conclusions are critical and concise. Figures and tables are explicit and they add to the message. Presentation logical and language are adequate.
Finally, the manuscript is well written and well presented, however, there are few issues that the authors should take into considerations. Below are some specific comments:
Statistics:
- The authors should describe in an additional paragraph of Part 2 “Materials and Methods” the statistical analyses carried out (tests used for each parameter studied, software, significance, ...).
Results:
- In figure 5E, authors must mention animal group on the representative immunoblot membrane. It may also be useful to present quantitative analyses for all groups (at least in supplementary data).
- It is the same remark for the figure 6. Since the 6-OHDA lesion is unilateral, inter-hemispheric compensation mechanisms probably occur and could help explain the differences between right and left. The results obtained for a control group (without injection of 6-OHDA) could remove this doubt.
Discussion:
- Authors should discuss the impact of compensation phenomena related to unilateral lesion of the SNpc in the interpretation of their results.
- It is not very clear how the authors explain the beneficial effect of iTBS on cellular and molecular changes. Authors select rats that will benefit from iTBS treatment or sham treatment when the DA neuronal cell death peaks (approximately 7 days after stereotaxic surgery, page 3 of 18, lines 131-134). The beneficial effects of iTBS in this case are probably not neuroprotective. It is probably necessary to weigh the conclusions and try to propose explanations on the mechanisms of action of iTBS, in particular to explain changes in TH positive fibers density in SNpc.
If mechanisms of compensation, plasticity seem to explain the changes in the striatum, it is more difficult to explain the changes in terms of the amount of DA neurons in the SNpc especially when the stimulations are performed when motor signs are detectable, so when cell death in the SNpc is well advanced. Authors must develop this point.
Then, I have some minor points:
1. Page 9 of 18, line 314 the right figure is probably the Figure 4 and not 3.